# The Impact of Self-Employment on the Health of Migrant Workers: Evidence from China Migrants Dynamic Survey

**DOI:** 10.3390/ijerph19105868

**Published:** 2022-05-11

**Authors:** Wanting Huang, Lei He, Hongxing Lan

**Affiliations:** College of Management, Sichuan Agricultural University, Chengdu 611130, China; 2020109002@stu.sicau.edu.cn (W.H.); 2021109009@stu.sicau.edu.cn (L.H.)

**Keywords:** migrants workers, health, self-employment, vulnerable group

## Abstract

Rural-to-urban migrant workers are at high risk of health inequalities in cities. Since labor is a central social determinant of health, this paper provided evidence on the health consequences of self-employment among mobile populations in developing countries. The cross-sectional data from the 2017 data of the China Migrants Dynamic Survey (CMDS) and the IV-Oprobit model are used to examine the effects of self-employment on health. The results showed that: (1) Self-employment was positively related to health; (2) among the self-employed, the health effects of opportunity self-employed are larger than those of necessity self-employed; (3) in the subsample analysis, the health effect of self-employment was greater for male and Han nationality migrant workers; (4) self-employment promotes health primarily through reducing manual labor, increasing flexibility time, job stability, financial rewards, and social integration directly or indirectly. Thus, focusing on improving the social security system, granting entrepreneurial subsidies, and optimizing the business environment mean boosting the positive effect of self-employment on economic development.

## 1. Introduction

Internal labor migration is a necessary livelihood strategy for millions of individuals and households throughout many developing counties [1]. Rural-to-urban migrant workers are known as the “mobility population” in China, emphasizing the characteristics of their transient and unstable social status in urban areas. In 2021, there were roughly 293 million migrant workers in China, accounting for 21% of China’s total workforce [2]. As a marginalized group, they suffer from poverty [3], poor housing quality [4,5], heavy physical labor, and dangerous working conditions [6,7], all of which result in severe health consequences. Furthermore, rural-to-urban migrant workers with a rural “hukou” are unable to obtain public health services in the cities where they work unless they change to an urban hukou, a process that can entail significant difficulties [8], so there are still many inequalities in the health of migrant workers [9].

Labor is a central social determinant of health [10], and occupational choice can influence job mobility and income trajectories [11,12] to affect health outcomes [13]. Supporting migrant workers to change their professional identity may be a pathway to improving health. In fact, as China’s different ownership economies continue to develop, shifting from employed to self-employed migrant workers has become an important trend in recent years [14], especially since the Chinese government proposed the strategy of “mass entrepreneurship and innovation” in 2014. According to data released by the National Health Care Commission, the proportion of self-employed migrant workers in China is over 25% [15,16], which includes solo self-employed who work for themselves in the non-agricultural sector, and employers who hire others. The impact of self-employment on the income [14,17,18] and life [19] of migrant workers has been widely studied, but the results of self-employment on health are still uncertain [20] because self-employment is associated with two opposite mechanisms on health. On the one hand, compared with wage work, self-employment can affect health by furnishing more room for health-enhancing behaviors as it allows flexible time arrangements to exercise, see a doctor in time, or develop healthy living habits [21,22]. On the other hand, self-employed may be unable to detach from work since they are responsible for their profits and losses, they will likely spend long hours on the job and reduce their own leisure time [23], making them more likely to be tired and exhibit insomnia [24]. Indeed, most of the literature in European countries also provided conflicting evidence: some studies show that entrepreneurship can improve health outcomes, for example, entrepreneurs in Germany are in better health [25], and self-employed workers in Portugal are about half as likely to be hospitalized as wage workers [20]; they are also associated with less mental health problems in Germany and Australia [26]. On the contrary, some studies have found that the health status of self-employed workers in EU countries is worse [24,27].

However, the data of these European countries or other developed countries are not enough to support conclusions on the health difference between self-employed workers and employees in developing countries. In recent years, the relationship between occupation and health has also attracted the attention of scholars in Asia [28,29]. However, they are based on samples of ordinary residents and pay little attention to samples of vulnerable groups such as Chinese migrant workers, and discussion on the transmission channel linking self-employment and health remains scant. In addition, most studies also ignore the different health effects of heterogeneous self-employment, and the potential missing variable bias and reverse causality from health to employment choice have not been fully dealt with. Therefore, extrapolated conclusions are very weak.

Based on the survey data of the China Migrants Dynamic Survey (CMDS) in 2017, this paper not only discusses the impact of self-employment on the health of migrant workers, but also divides self-employment into necessity self-employment and opportunity self-employment, supplementing the impact of heterogeneous self-employment on rural migrants’ health. In addition, the transmission channel linking self-employment and health is discussed, and the instrumental variable is introduced as much as possible to solve the endogenous problem.

Compared with previous studies, the marginal contributions of this article are as follows: Firstly, based on a large sample of microdata (2017 China Migrants Dynamic Survey data), this paper focuses on rural migrants and separates self-employment from informal employment and studies the health and transmission channel from the perspective of self-employment, which expands the framework of factors influencing migrants’ health. Secondly, based on active choice and passive choice of self-employment, this paper distinguishes the impact of different self-employed migrant workers on health to avoid covering up some structural problems. Thirdly, the introduction of instrumental variables makes up for the lack of endogenous discussion in the existing literature. Based on the above analysis, this paper responds to problems in the health intervention path of mobile populations and provides policy references for further improving the health level of migrants and for public health under the background of the growth of novel flexible work formats.

## 2. Theoretical Analysis and Hypotheses

### 2.1. Direct Impact of Self-Employment on the Health of Migrant Workers

The most direct way in which self-employment affects the health of migrant workers is through the nature of the occupation. Firstly, compared to the labor-intensive industries such as construction and manufacturing where a large portion of employed migrant workers is located, self-employed migrant workers are mainly concentrated in the wholesale and retail industries, restaurants and lodging, and other living services [30], which allows them to reduce manual labor and contributes to their health. Secondly, the health effects of self-employment mainly stem from the incentives of an independent work/life and “being your own boss” [31,32]; they are more satisfied with their jobs because of the greater autonomy of independent work, as well as more flexible working hours and greater job security [33,34,35]. Accordingly, this paper proposes Hypothesis 1:

**Hypothesis** **1.**
*Self-employment activities have a positive impact on the health of migrant workers as self-employment offers less manual labor, more flexible time, and higher job security.*


### 2.2. Indirect Impact of Self-Employment on the Health of Migrant Workers

Compared to being employed, self-employment has a higher income premium [17], and increased income levels may drive potential health consumption and thus promote their health [36]. In addition, migrant workers generally face a lack of urban social security, and wage-earning migrant workers are often at risk of inadequate health care affordability due to both social security and income levels, but self-employed individuals are also more able to finance their health care [37]. Thus, self-employment may have an impact on health by increasing migrant workers’ income and ability to pay for health care.

In addition, existing research suggests that self-employed people have an advantage over employed people in terms of social capital in the “dinner network” and “New Year’s greeting network” [38]. As China is an “acquaintance society”, the specific network can not only significantly improve their business performance [39], but also play an informal guarantee role against the impact of serious diseases [40] and help them to mobilize social resources, receive mutual assistance, and accumulate information about health [41,42]. Moreover, in studies of mobile populations, identity is regarded as an important indicator of social integration at the psychological level and as a higher-order form of social integration [43]. The classical social identity theory defines identity as “the recognition by an individual of his or her membership in a social group and the value and emotional meaning attached to that membership” [44]. The identity of the mobile population emphasizes that its members can truly be integrated into the local society if they have completed the transformation of their identity psychologically [45]. The household registration system is the root cause of the identity dilemma of the migrant population [46], as well as the rejection by the flowing society [43], while frequent contact with local people and active participation in local organizations will alleviate the identity crisis [47]. Compared with employed migrant workers, the rich social network and social interaction of self-employed migrant workers make their sense of belonging and self-perceptions enhanced and their identity is further strengthened [48], which has been shown to be very effective in boosting health [49,50]. Accordingly, this paper proposes Hypothesis 2: 

**Hypothesis** **2.**
*Self-employment may indirectly affect the health of migrant workers by affecting financial rewards and social integration.*


In conclusion, the theoretical analysis framework of this paper is shown in Figure 1.

Furthermore, differences in entrepreneurial motivation, gender, and ethnicity among self-employed migrant workers may lead to differences in health.

Firstly, whether one becomes self-employed to escape unemployment or to pursue one’s business ideas could have implications for health as their life goals, self-fulfillment, and financial rewards differ. For migrant workers who actively become self-employed (opportunity self-employed), they tend to be more skilled, have the flexibility to choose their working hours and methods, and enjoy their work more, which results in lower mental burden and good health status [51]. On the contrary, those who cannot find a formal job and passively become self-employed (necessity self-employed) are often forced to work in lower-end jobs and work longer hours due to the pressure of life and being limited by their skills [52], making them experience poor health [53] compared with opportunity self-employed. Furthermore, the opportunity self-employed start businesses mainly to increase (rather than maintain) their income or to become independent [54,55], which also helps improve their ability to invest in health.

Secondly, gender discrimination has long existed in the labor market, and female migrant workers are a vulnerable group of migrant workers [56,57]. Engaging in self-employment helps female migrant workers get rid of heavy manual labor. However, the labor intensity of male migrant workers in the labor market is higher than that of female migrant workers, and thus the marginal effect of self-employment on health is higher for male migrant workers.

Thirdly, China is a multi-ethnic country and the ethnic minority migrant workers face the dual challenges of urban-rural mobility and cross-ethnic interactions [58]. They are usually grouped by ethnicity and religion, and their social structure is “involuted” and isolated from the mainstream society on the mainland [59]. As a result, their being self-employed is more costly, which has a smaller effect on their health relative to Han nationality migrant workers. Accordingly, this paper proposes Hypothesis 3a, Hypothesis 3b and Hypothesis 3c:

**Hypothesis** **3a.**
*Opportunity self-employed migrant workers are expected to experience greater health benefits than necessity self-employed migrant workers.*


**Hypothesis** **3b.**
*Male migrant workers are expected to experience more health benefits than female migrant workers.*


**Hypothesis** **3c.**
*Han nationality migrant workers are expected to experience more health benefits than ethnic minority migrant workers.*


## 3. Materials and Methods

### 3.1. Data Sources

The China Migrants Dynamic Survey (CMDS) was conducted by the National Health and Family Planning Commission (NHFPC) covering 31 provinces (districts, cities) and the Xinjiang Production and Construction Corps in mainland China. The NHFPC began the survey in 2009 to track the living conditions of migrants aged over 15 years, residing in the host city for over one month, and without local hukou; this survey is the largest and most representative mobile population database in China. The sample of the 2017 CMDS was based on the 2016 annual report on the population of migrants and a total of 169,989 samples were collected by stratified sampling and the multi-stage probability proportionate to size sampling (PPS) method. The PPS method means that the probability of each unit being selected in each sampling is proportional to the size of the unit. Therefore, it is characterized by the probability that a large part of the overall content will be sampled, which can improve the representativeness of the sample. In this survey, stratification was carried out according to provinces (autonomous regions and municipalities), cities or districts, townships (or neighborhoods), and villages (or residential committees); in each stratum, sampling was carried out according to the PPS method, and the sample mobile population was finally drawn. The auxiliary information on population size was used in each stratum and each stage to reduce the sampling error.

To screen out migrant workers with different employment statuses, this paper makes the following provisions: (1) It is required to be a rural household with rural hukou; (2) the age range is 18–60, which is the most common practice and can be easily compared with other studies; (3) the reason for inflow is to work or do business in the city and having a non-farm job in the previous week. (4) selects only those who are currently employed as employees or self-employed; (5) removes some missing values, such as education, income, medical insurance, etc. The number of final observations in this paper is 96,792. Among them, the sample of employees is 55,724, accounting for 57.57%, while the sample of self-employed migrant workers is 41,068, accounting for 42.43%.

### 3.2. Variable Definition

#### 3.2.1. Explained Variables

Health status is the core explanatory variable in this study. The measurement of health has been one of the difficulties in health economics, and the existing literature mainly uses self-assessment of health [60,61]. The greatest advantage of the indicator is that it is easy to obtain and provides a more comprehensive measure of an individual’s overall health status. Because of this, this study also used the response to “How healthy are you” to classify health status as “very poor”, “poor”, “average” and “good”, and assigns values of 1–4, respectively. 

#### 3.2.2. Explanatory Variables

Referring to the research of Tervo [62] and Zhu [63], people are defined as self-employed workers if they are responsible for the profit and loss of the business and are paid from self-employment accounts. Based on this, combined with the employment status survey in the CMDS questionnaire, this paper defined “employers” and “self-employed workers” as “self-employed” with a value of 1, and “employees” as “wage workers” with a value of 0. Moreover, depending on the different motivations and stages of development of self-employment, “employers” were defined as “opportunity self-employed”, which also means they hired others, while “self-employed workers” who are solo entrepreneurs are considered as “necessity self-employed” [64,65].

#### 3.2.3. Other Variables

In the mechanism analysis, the responses to the questions “the industry you work in”, “whether or not lack of time to see a doctor” and “whether or not face difficulties of unstable work “were selected to verify the direct effect of self-employment on individual health. The “wage income” and the answers for “whether or not lack of money to see a doctor “were selected as the measures of financial return, and the answers for “Whether the person interacts most is local residents” and “whether or not I identify myself as a local” were selected as the as indicators of social integration, which were used to test the indirect mechanism of self-employment on the health of migrant workers. Moreover, in order to measure more accurately and avoid the problem of omitted variables, a series of control variables were also included, such as personal characteristics, work characteristics, and regional characteristics. All variables are shown in Table 1.

### 3.3. Methods

In this paper, the self-rated health is an ordinal variable with a value from 1 to 4, and the Oprobit model is used to examine the effect of self-employment on migrant workers’ health:(1)$$Health_{i}=\alpha SE_{i}+\beta X_{i}+\varepsilon_{i}\;$$
where i denotes the ith sample, and the explanatory variable Health denotes the health status of individual i. The core explanatory variable SE indicates whether an individual is self-employed, with a value of 1 for “self-employed” and 0 otherwise; a series of control variables $X_{i}$ mainly includes personal, work, and regional characteristics. $\varepsilon_{i}$ is the error term.

It has been shown that healthy people are more likely to be self-employed because healthier people can focus on exploring business opportunities [66] and have access to critical financing for business activities [67]. Therefore, there are reverse causality or omitted variables between self-employment and health. We tried to adopt the average rate of self-employment in each subgroup as an instrumental variable to overcome the endogeneity problem. The main reason is that migrant workers in Chinese cities often live together, and are lacking information and have low income; their interactions and behaviors refer to people who are similar to themselves or in the same class, which form the “peer effect” [68]; the higher the self-employment rate in a region, the more people around an individual may choose to engage in self-employment while being almost independent of individual health. 

The IV-probit model is set as follows:(2)$$SE_{i}=\delta_{i}Z_{i}+\theta_{i}X_{i}+\mu_{1}\;$$
(3)$$Health_{i}=\emptyset_{i}Z_{i}+\rho_{i}SE_{i}+\omega_{i}X_{i}+\tau_{i}\;$$
where $Z_{i\;}$ represents the tool variables of this article, and $\mu_{1}$ and $\tau_{i}$ are random disturbance terms satisfying cov ($\mu_{1}$, $\tau_{i}$) ≠ 0.

Migrant workers who choose self-employment may have higher business skills and risk awareness due to their resource endowments, while migrant workers who do not become self-employed may be weak in these aspects, or they may not be willing to give up their current career. Ordinary regression can only observe the health status of migrant workers with self-employment behavior, but for non-self-employed migrant workers, the impact on their health if they become self-employed cannot be observed. Therefore, there is a “self selection bias”. This paper uses the Propensity Score Matching (PSM) method to solve the problem by obtaining a consistent estimated Average Treatment Effects on Treated (*ATT*) by comparing it with the observed self-employment group. The specific calculation formula of *ATT* is:(4)$$ATT=E\left[Health_{1i}|A_{i}=1\right]-E\left[Health_{0i}|A_{i}=1\right]\;$$

$Health_{1i}\;$represents the health status of migrant workers who are self-employed, and $Health_{0i}\;$represents the health status of migrant workers who are not self-employed, which is constructed through the counterfactual framework.

Finally, this paper constructs an intermediary effect model. Based on the above, we estimate the impact of self-employment on mediating variables and then test the impact of self-employment and mediating variables on health.

## 4. Results

### 4.1. Descriptive Results

Table 2 shows the results of the descriptive statistical analysis of the relevant variables:

(1)In the whole sample, about 42.43% of migrant workers engaged in self-employment had the group characteristics of higher age, higher proportion of male and Han nationality, and lower education level compared with wage workers. Specifically, the average age of the self-employed was-37.939 years, about 3.5 years significantly older than that of wage workers. Below the age of 35, there were more wage workers, while after the age of 35, the situation changed and the proportion of self-employed migrant workers gradually exceeded that of wage workers. In fact, this distribution characteristic is consistent with reality: on the one hand, workers tend to choose to be employed when they are young to accumulate more human and monetary capital, and then turn to being self-employed when they are middle-aged. On the other hand, the greater family support pressures faced by migrant workers in their middle age may also motivate or force individuals to engage in self-employment activities. The proportion of male migrant workers who were self-employed was roughly 1.3 percentage points larger than that of female, and the proportion of Han nationality migrant workers who were self-employed was over 90%. The average educational attainment of self-employed was 9.258 years, while the wage for workers was 10.205.(2)In terms of work characteristics, the average monthly income of self-employed was CNY 1359.80 higher than that of wage workers, but at the same time, the self-employed worked 11 h more per week. This suggests that migrant workers are self-employed out of economic rationality, but higher income is accompanied by more working hours. In addition, self-employed migrant workers accounted for a higher proportion of urban employees’ medical insurance and health records.(3)Moreover, in terms of regional characteristics, the proportion of self-employed migrant workers in the central and western regions was significantly higher than that in the eastern and northeastern regions. It shows that the areas with a relatively inactive economy are likely to increase the proportion of self-employment. A total of 75.3% of self-employed migrant workers were mainly concentrated in the living service industry, while the secondary and productive service industries accounted for 21.3% and 3.4%, respectively.(4)The mean value of health of the self-employed was 3.823, indicating that the self-employed migrant workers health is above the average level, which was lower than that of wage workers. At the same time, compared to employed migrant workers, self-employed migrant workers were significantly more represented in non-manual labor industries, had significantly fewer difficulties with job instability, and had significantly higher wages, social capital, and psychological identity than employed migrant workers. However, there were no significant difference between the two groups in terms of time flexibility and medical payment ability.

### 4.2. Empirical Results

#### 4.2.1. Influence of Self-Employment on Health

Table 3 shows the effect of self-employment on migrant workers’ self-rated health using an ordered probit model. In model 1, we only include self-employment and regional variables, which produces the simple association between self-employment and health; the coefficient is negative at a significance level of 1%. In model 2, we add the demographic control variables and find that health is significantly positively associated with self-employment (*p* < 0.01). In model 3, control variables such as income or working hours and insurance continue to be included, and the coefficient becomes larger (*p* < 0.01). These results also indicate that older, women, and ethnic minority workers exhibit lower levels of health. Furthermore, working hours (*p* < 0.01) and medical insurance (*p* < 0.01) are negatively correlated with health, while income level (*p* < 0.01) and health record (*p* < 0.01) coefficients are significantly positive.

In order to solve the endogenous problem of reverse causality, this paper uses the IV-Oprobit model for further estimation, and the results are shown in Model 4. Firstly, although it is not reported in Model 4, the coefficient between regional self-employment rate and migrant workers’ self-employment behavior is 0.742 and significant at the 1% level, while the first-stage F-value is greater than 10. Secondly, the lnsig_2 value is −0.912 and the two-stage estimation of the model is significant and passes the atanhrho_12 test, which proves that the method is better than Oprobit estimation and the instrumental variable is reasonably chosen. Finally, the result indicates that self-employment has a positive impact on health, which tentatively confirms Hypothesis 1.

Further, the Propensity Score Matching (PSM) method can effectively address the problem of sample self-selection bias and can provide robust support for the baseline regression results. The most commonly used K-value Neighbor (K = 4), Caliper and Radius (cal = 0.01), and Kernel and Local Linear matching methods were used for estimation in this study. The results of the equilibrium test based on the nearest neighbor matching method show the standard deviations of the matched samples were all reduced to less than 10%, which indicates that the differences in sample characteristics were eliminated to a large extent. The results in Table 4 show the symbols and significance of the three matchings are the same, and the average treated effect (*ATT*) of self-employment on migrant workers’ health is about 0.025, which is similar to the estimation result of the above model. Therefore, the conclusion that self-employment significantly improves the health level of migrant workers is robust.

In addition, we further adopted objective health indicators of “past year prevalence”, and the IV-probit model was used for robustness testing. The explained variables in models 1 to 6 are “diarrhea in the past year”, “fever in the past year”, “skin rash in the past year”, “jaundice in the past year”, “conjunctivitis in the past year” and “cold in the past year” (0 = no), and the explained variable for model 7 was “being sick in the past year” (0 = no) to express the overall objective health status of migrant workers. The analysis in Table 5 shows that the coefficients of the six short-term health indicators in models 1–6 are mostly negative, but the indicator “jaundice in the past year” is not statistically significant, which shows that the effect of self-employment on the prevalence of different symptoms is negative in general. The coefficients of the “sick” of model 7 are also negative at the 1% level of significance, which further validates the conclusion that self-employment is beneficial in reducing the prevalence of diseases and has a positive impact on health among migrant workers. 

#### 4.2.2. Influence of Heterogeneous Forms of Self-Employment on Health

Based on the examination of different self-employment categories, the explanatory variables were classified as opportunity self-employment and necessity self-employment. The results are shown in Table 6. In general, the results in Model 1 show opportunity self-employed workers experience higher rated-reported health compared with necessity self-employed workers. However, necessity self-employment reduces more of the likelihood of sickness of migrant workers in Model 2. Thus, Hypothesis 3a is not fully verified.

#### 4.2.3. Influence of Self-Employment on Health by Gender and Nationality

Regarding other heterogeneous effects, we find that the positive effect of self-employment on health is larger among men and Han nationality workers (Table 7). However, there is a possibility that the effective sample size available for the ethnic minority information of only 8615 results in a loss of efficiency in the regression analysis, which verifies Hypothesis 3b and Hypothesis 3c.

### 4.3. Mechanism Analysis

#### 4.3.1. Direct Mechanism Analysis

The previous section discussed the mechanisms of self-employment affecting the health of migrant workers, the direct mechanism of which is through the nature of the work. The results of the direct mechanism analysis (Table 8) show that if migrant workers are self-employed, they will engage in less manual labor and will not delay medical appointments due to flexible time, as well as feel less unstable about their jobs. Therefore, reducing manual labor and increasing flexible time and working stability is a direct way for self-employment to enhance migrant workers’ health, which verifies Hypothesis 1.

#### 4.3.2. Indirect Mechanism Analysis

Self-employment may also affect individual health by influencing financial return and social integration; the results of indirect mechanism analysis are shown in Table 9. In terms of financial return, self-employment decisions have significant positive effects on wage, and the coefficient of self-employment on health decreases after adding wage to the health equation compared to the coefficient (0.081) without controlling total revenue. After that, if the control variables are as in Table 3, the results show that the self-employed have a higher ability to pay for health care, and the coefficient (0.056) of self-employment on health is still significantly positive but smaller than the coefficient (0.063) of Model 3 in Table 3, indicating that wage and medical payment ability weaken the effects of self-employment on migrant workers’ health. Similarly, the two mediating variables of social capital and psychological integration have the same effect, which indicates that economic reward and social integration are indirect channels through which self-employment enhances migrant workers’ health, which verifies the Hypothesis 2.

## 5. Discussion

In this study, based on a large sample size survey, 96,792 Chinese migrant workers were selected to study the impact of self-employment on health. This study not only focused on the impact of self-employment on health, but also investigated the health effects of different forms of self-employment, and tested the direct and indirect paths of the impact of self-employment on health. We constructed an ordered probit model to overcome the endogenous problem of variable selection; the PSM method was used to deal with the sample self-selection bias, and the intermediary model was used as the mechanism test, which is more comprehensive. As the largest developing country with the largest number of migrant workers in the world, the research results for China have strong practical significance. 

This study found that self-employment has a significant positive impact on health. This is consistent with the results of Goncalves et al. [20], who found that the likelihood of hospital admission of self-employed individuals is about half that of wage workers in Portugal. On the other hand, our results are consistent with Rietveld et al. [67] and Lee et al. [69], who found that engaging in self-employment is bad for one’s health. The former believes that the conclusion is explained by a selection effect, in which healthier individuals self-select into self-employment, while the latter selected the elderly aged 55 to 84 to study their physical health status after switching from retirement to self-employed, which leads to the difference from the results of this paper.

We also found that the health effect of opportunity self-employment is higher than that of necessity self-employment. The result is inconsistent with Nikolova [70], who pointed out that necessity entrepreneurs experience improvements in their mental but not physical health, while opportunity entrepreneurship leads to both physical and mental health gains. The difference in the research results may be due to the different initial employment statuses of the sample. Nikolova’s study focused on the health effects of switches from unemployment to self-employment (necessity employment) and transitions from regular sector to self-employment, while the reference objects of our study are employed migrant workers. In addition, when we replaced the explanatory variables with the likelihood of sickness and morbidity rate, the necessity self-employment has a greater weakening effect on those two indicators. The result is consistent with the evidence from Blanchflower [23] and Hessels et al. [26], who also found that the self-employed with employees experience higher stress, exhaustion, and depression than regular employees and solo entrepreneurs; because self-employed with employees may have to act as managers, recruiters, and accountants, these high job demands may increase exhaustion [71]. Therefore, the marginal utility of necessity self-employment on migrant workers’ morbidity is greater. This suggests that as Chinese migrant workers’ urban integration deepened, self-employment became a choice based on comparative advantage for migrant workers; despite some pressure, the self-employment activities had also become a kind of “decent employment”.

Whether male or female migrant workers, self-employment is conducive to their health, which also shows that self-employment activities allow Chinese women to break the division of labor in the family mode of “the man goes out to work while the woman looks after the house” division of labor. Therefore, encouraging self-employment is a way to eliminate gender discrimination in the labor force [72]. However, the health effect of self-employment on ethnic groups is not obvious, which may be ascribed to weak resource endowments and the “involution” of social capital, showing that self-employment is still a vulnerable area for ethnic minorities.

Getting rid of heavy manual labor, having more flexible time for health management, and keeping workers in a stable state of work are the benefits missing from the employment of traditional migrant workers, but the working nature of self-employment activities makes up for these disadvantages. In addition, higher economic returns and more “localized” social capital brought by self-employment activities can indirectly improve the health level of migrant workers. The analysis of these mechanisms shows that the impact of self-employment on the health of migrant workers has not only monetary returns but also non-monetary returns.

In addition, this study has several deficiencies that can be addressed in future studies, which are as follows: We selected the cross-sectional data of the 2017 CMDS as the research sample for this paper. However, the influence of self-employment on migrant workers health is a dynamic process. Thus, future research could use panel data to further expand and verify the relationship in greater detail.Studying the relationship between self-employment and health also involves many missing variables, such as previous unhealthy habits, illness, original industry choices, risk appetite and perseverance, and even genes [73,74]; if these variables can be controlled, the processing effect will be cleaner, but it falls outside the scope of this paper to discuss them due to the availability of data and samples.The choice of instrumental variables can be further deliberated on. Previous studies have used the number of self-employed members in the household, immigration variable status, and the presence of insurance for children as instrumental variables [37], but the above data were not collected in the database we selected. The regional registered unemployment rates can also be used [28], but the classical literature confirms that a 1% increase in the U.S. unemployment rate will reduce overall mortality by 0.5% [75]. Therefore, based on data availability and “relative safety”, this paper only constructs a higher latitude district-level self-employment rate as an instrumental variable, which is, of course, a less than perfect and skillful approach.

## 6. Conclusions and Implications

Based on the survey data from the 2017 CMDS, the IV-Oprobit model was mainly used to quantitatively study the impact of self-employment and its heterogeneity on the health of migrant workers, and the following main conclusions were drawn:(1)The health status of self-employed migrant workers is better than that of wage workers, and this relationship is more pronounced after the introduction of instrumental variables, indicating that self-employed migrant workers experience more health benefits; the conclusions are still robust after replacing health indicators in the analysis model.(2)Most self-employed migrant workers in China are still necessity self-employed (35.40% of the total self-employed sample) and compared with wage workers, opportunity self-employment and necessity self-employment both have a significant positive effect on health, but opportunity self-employment has a higher effect on self-rated health and a lower effect on the likelihood of sickness and the morbidity rate.(3)Self-employed women are free from labor market discrimination and constraints, but the health effects of self-employment are not yet evident for ethnic minorities.(4)Self-employment not only directly promotes the health of migrant workers through less physical labor, more flexible time, and more stable nature of work, but also indirectly promotes the health of migrant workers through economic return and social integration.

With the rapid economic growth in China, self-employment has injected new dynamics into economic development. However, the self-employed are still a vulnerable group in the labor market compared to the wage employment group, which is specifically reflected in the lack of social security and financing environment. The above findings have the following obvious policy implications:(1)Health should be taken into account as an important objective in employment policy formulation, and a social security system should be established to match labor mobility, focusing on solving the problems of urban-rural “fragmentation” of basic social insurance, inadequate transfer connections, and high transfer costs so as to protect the health rights and interests of self-employed migrant workers.(2)Opportunistic self-employment can create more jobs and is an important source of economic vitality. Based on the fact that opportunity self-employment depends more on the optimization of external conditions, it is important to promote the improvement of the business environment, alleviate financial constraints, and provide training on business laws to relieve the work pressure of self-employed workers and improve their health level. For the necessity self-employment, we can provide more convenient services in terms of employment skills and job security to improve their health benefits.(3)The importance of social security subsidies for self-employed women, cultivating women’s entrepreneurship or entrepreneurial skills, and improving the childcare welfare system to eliminate the health gender disparity of self-employment should be emphasized. Employment support policies for ethnic minorities need to be further improved, such as strengthening vocational skills education and legislating against employment discrimination. In addition, it is necessary to improve the income of migrant workers through multiple channels, pay attention to the cultivation of migrant workers’ social ability, and enhance their psychological identity with urban integration.

## Figures and Tables

**Figure 1 ijerph-19-05868-f001:**
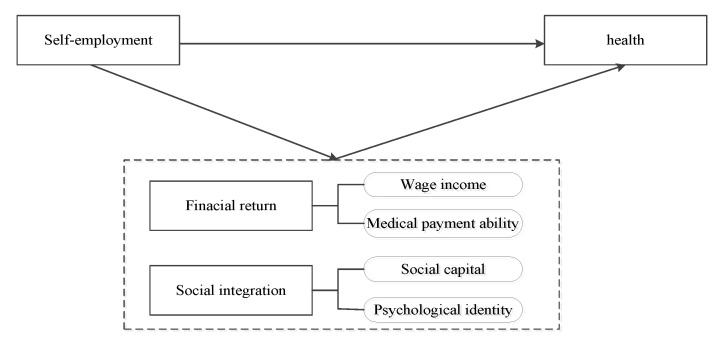
Theoretical analysis of self-employment affecting health.

**Table 1 ijerph-19-05868-t001:** Variable selection, definition, and assignment.

Variable Classes	Variable Name	Variable Meaning and Assignment
Explained variables	Self-rated health	Self-rated health: 1 = very poor (unable to take care of myself); 2 = poor (unhealthy but able to take care of myself); 3 = average; 4 = good.
Explanatory variables	Self-employment	Whether to engage in self-employed activities: 1 = yes; 0 = no
	Opportunity self-employment	Whether or not hire other worker:1 = yes; 0 = no
	Necessity self-employed	Whether it is solo entrepreneurs: 1 = yes; 0 = no
Control variables	Age	Age in years
	Gender	1 = male; 0 = female
	Nationality	1 = Han nationality; 0 = others
	Education	Education in years
	Work_time	Hours worked last week
	Income	The logarithm of monthly average total income
	Medical Insurance	Availability of Urban Employee Medical Insurance: 1 = yes; 0 = no
	Health record	Availability of health records: 1 = yes; 0 = no
	industry	1 = productive services; 2 = secondary industry; 3 = living services
	region	1 = East; 2 = Central; 3 = West; 4 = Northeast.
Other variables	Manual labor	Whether or not in a labor-intensive industry: 1 = yes; 0 = no
	Flexible time	Whether or not lack of time to see a doctor:1 = no; 0 = yes
	Working stability	Whether or not face difficulties of unstable work: 1 = no; 0 = yes
	Wage	The logarithm of last month’s total income
	Medical payment ability	Whether or not lack of money to see a doctor: 1 = no; 0 = yes
	Social capital	Whether the person interacts most is local residents: 1 = yes; 0 = no
	Psychological identity	Whether they identify themselves as local people: 1 = not at all, 2 = not; 3 = basically; 4 = fully

**Table 2 ijerph-19-05868-t002:** Descriptive statistics.

Index	Self-Employed	Wage Workers	Difference between Group
Health	3.823	3.845	0.0208 ***
Age	37.939	34.431	−3.508 ***
Age groups			
18–24	0.047	0.150	0.104 ***
25–34	0.341	0.408	0.067 ***
35–44	0.354	0.257	−0.096 ***
45–54	0.221	0.154	−0.067 ***
54–60	0.037	0.030	−0.008 ***
Gender	0.598	0.585	−0.013 ***
Nationality	0.920	0.904	−0.015 ***
Education	9.258	10.205	0.947 ***
Work_time	65.211	53.732	−11.408 ***
Income	7796.68	6436.976	−1359.80 ***
Medical insure	0.039	0.023	−0.015 ***
Health record	0.271	0.238	−0.033 ***
Industry groups			
Productive services	0.034	0.080	0.047 ***
Secondary industry	0.213	0.516	0.388 ***
Living services	0.753	0.403	−0.035 ***
Region			
East	0.313	0.515	0.203 ***
Central	0.245	0.149	−0.096 ***
West	0.398	0.276	−0.121 ***
Northeast	0.045	0.060	0.015 ***
Manual labor	0.237	0.567	0.330 ***
Flexible time	0.966	0.967	−0.001
Working stability	0.806	0.774	−0.032 ***
Wage	4556.779	3859.6	−697.180 ***
Medical payment ability	0.987	0.986	−0.001
social capital	0.311	0.251	−0.006 ***
Psychological identity	2.968	2.829	−0.139 ***
Observations	41,068	55,724	

The second and third columns in the table are the mean values; *** represent significance at 1% levels, respectively.

**Table 3 ijerph-19-05868-t003:** Effect of self-employment on health.

Variables	Oprobit			IV-Oprobit
	Model 1	Model 2	Model 3	Model 4
Self_employment	−0.041 ***	0.056 ***	0.063 ***	0.230 ***
	(0.010)	(0.010)	(0.012)	(0.037)
Age groups (base:18–24)				
25–34		−0.215 ***	−0.252 ***	−0.274 ***
		(0.021)	(0.021)	(0.021)
35–44		−0.446 ***	−0.483 ***	−0.516 ***
		(0.021)	(0.021)	(0.022)
45–54		−0.727 ***	−0.759 ***	−0.790 ***
		(0.022)	(0.022)	(0.023)
55–60		−0.990	−1.012 ***	−1.038 ***
		(0.030)	(0.030)	(0.030)
Gender		0.087 ***	0.093 ***	0.088 ***
		(0.010)	(0.011)	(0.010)
Nationality		0.070 ***	0.068 ***	0.061 ***
		(0.017)	(0.017)	(0.018)
education		0.026 ***	0.022 ***	0.024 ***
		(0.002)	(0.002)	(0.002)
Work_time			−0.002 ***	−0.003 ***
			(0.000)	(0.000)
lncome			0.095 ***	0.079 ***
			(0.010)	(0.010)
Medical insurance			−0.086 ***	−0.096 ***
			(0.028)	(0.028)
Health record			0.097 ***	0.096 ***
			(0.012)	(0.012)
Industry (base: productive services)				
Secondary industry			0.047 *	0.050 *
			(0.022)	(0.022)
Living services			0.045 *	−0.000
			(0.022)	(0.024)
Region (base: East)				
Middle	−0.196 ***	−0.219 ***	−0.211 ***	−0.237 ***
	(0.013)	(0.014)	(0.014)	(0.015)
West	−0.153 ***	−0.133 ***	−0.115 ***	−0.145 ***
	(0.011)	(0.012)	(0.012)	(0.014)
Northeast	−0.185	−0.137 ***	−0.110 ***	−0.114 ***
	(0.022)	(0.022)	(0.022)	(0.022)
Pseudo R^2^	0.004	0.043	0.045	
Lnsig_2				−0.912 ***
				(0.002)
atanhrho_12				−0.074 ***
				(0.016)
LR chi^2^/Wald chi^2^	357.20 ***	3887.78 ***	4124.94 ***	53,636.20 ***
Log likelihood	−45,269.725	−43,504.435	−43,385.855	−92,405.983
Observations	96,792	96,792	96,792	96,792

*, and *** represent significance at 10 and 1% levels, respectively; robust standard errors are in parentheses.

**Table 4 ijerph-19-05868-t004:** The results of propensity score matching.

	Sample	Treated	Controls	*ATT*	Standar Error	T-Value
Health	Unmatched	3.824	3.845	−0.021	0.003	−7.89
	K-value Neighbor (K = 4)	3.824	3.797	0.027	0.004	5.63
	Caliper and Radius (cal = 0.01)	3.824	3.799	0.025	0.005	4.52
	Kernel and Local Linear	3.825	3.801	0.024	0.007	3.90

**Table 5 ijerph-19-05868-t005:** Robustness test results.

Variables	Mode 1	Model 2	Model 3	Model 4	Model 5	Model 6	Model 7
	Diarrhea	Fever	Skin Rashes	Jaundice	Conjunctivitis	Cold	Sick
Self-employment	−0.094 **	−0.139 ***	−0.381 ***	0.171	−0.199 ***	−0.208 ***	−0.535 ***
	(0.039)	(0.040)	(0.056)	(0.180)	(0.068)	(0.031)	(0.030)
Control variables	Control	Control	Control	Control	Control	Control	Control
Constant	−2.031 ***	−1.501 ***	−2.884 ***	−2.000 ***	−2.756 ***	−0.993 ***	−1.745 ***
	(0.105)	(0.110)	(0.151)	(0.489)	(0.184)	(0.084)	(0.082)
Wald chi^2^	684.72 ***	354.41 ***	195.39 ***	65.34 ***	99.16 ***	1055.37 ***	1786.20 ***
Observations	96,792	96,792	96,792	96,792	96,792	96,792	96,792

IV-probit model is used in the Table 5; the control variables are the same as in Table 3; the “non-self-employment” is the reference group; **, *** represent significance at 5 and 1% levels; robust standard errors are in parentheses.

**Table 6 ijerph-19-05868-t006:** Influence of different forms of self-employment on health.

Variables	Oprobit	Probit
	Model 1 Health	Model 2 Likelihood of Sick
Opportunity self-employed	0.069 ***	−0.139 ***
	(0.023)	(0.019)
Necessity self-employed	0.062 ***	−0.186 ***
	(0.012)	(0.010)
Control variable	control	control
Constant		−1.205 ***
		(0.073)
Pseudo R^2^/R-squared	0.045	0.013
LR chi^2^/wald chi^2^	4125.03 ***	1757.83 ***
Log likelihood	−43,385.808	−66,148.823
Observations	96,792	96,792

The control variables are the same as in Table 3; the “non-self-employment” is the reference group; *** represent significance at 1% levels; robust standard errors are in parentheses.

**Table 7 ijerph-19-05868-t007:** Heterogeneous effects of self-employment on the likelihood of hospitalization.

Variables	Mode1	Model 2	Model 3	Model 4
	Men	Women	Han Nationality	Ethnic Minority
Self-employment	0.264 ***	0.183 ***	0.258 ***	−0.072
	(0.051)	(0.054)	(0.039)	(0.118)
Control variable	control	control	control	control
Wald chi^2^	30,391.51 ***	24,105.80 ***	49,129.56 ***	4680.29 ***
Log likelihood	−54,638.587	−37,477.517	−83,939.327	−8366.088
Observations	57,176	39,616	88,177	8615

IV-Oprobit model is used in Table 7; the control variables are the same as in Table 3; *** represent significance at 1% levels; robust standard errors are in parentheses.

**Table 8 ijerph-19-05868-t008:** The results of direct mechanism.

Variables	Model 1	Model 2	Model 3
	Manual Labor	Flexible Time	Working Stability
Self-employment	−0.458 ***	0.111 ***	0.241 ***
	(0.032)	(0.017)	(0.011)
Constant	1.504 ***	2.496 ***	−1.872 ***
	(0.239)	(0.141)	(0.085)
Control variable	control	control	control
Constant	1.504 ***	2.496 ***	−1.872 ***
	(0.239)	(0.141)	(0.085)
Pseudo R^2^	0.640	0.014	0.055
LR chi^2^	18,043.75 ***	398.73 ***	5531.79 ***
Log likelihood	−5085.095	−14,069.308	−47,295.463
Control variable	control	control	control
Observations	96,792	96,792	96,792

Probit model is used in Table 8; the control variables are the same as in Table 3; *** represent significance at 1% levels; robust standard errors are in parentheses.

**Table 9 ijerph-19-05868-t009:** The results of indirect mechanism.

Variables	Financial Return				Social Integration			
	Model 1		Model 2		Model3		Model 4	
	OLS	Oprobit	Probit	Oprobit	Probit	Oprobit	Oprobit	Oprobit
	Wage	Health	Medical Payment Ability	Health	Social Capital	Health	Psychological Identity	Health
Self-employment	0.071 ***	0.079 ***	0.181 ***	0.056 ***	0.109 ***	0.060 ***	0.077 ***	0.097 ***
	(0.007)	(0.011)	(0.026)	(0.012)	(0.010)	(0.012)	(0.008)	(0.007)
Wage		0.058 ***						
		(0.004)						
Medical payment ability				0.705 ***				
				(0.033)				
Social capital						0.081 ***		
						(0.012)		
Psychological identity								0.058 ***
								(0.012)
Constant	7.824 ***		0.254 ***		−1.652 ***			
	(0.025)		(0.200)		(0.078)			
Control variable	control	control	control	control	control	control	control	control
Adj R^2^/Pseudo R^2^	0.058	0.0460	0.057	0.050	0.067	0.046	0.038	0.048
Observations	96,792	96,792	96,792	96,792	96,792	96,792	96,792	96,792

The control variables are the same as in Table 3; *** represent significance at 1% levels, respectively; robust standard errors are in parentheses.

## Data Availability

Restrictions apply to the availability of these data. Data were obtained from the Migrant Population Service Center, National Health Commission P.R. China and are available from Wanting Huang with the permission of the center.

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
