# Peer review of "The Impact of Self-Employment on the Health of Migrant Workers: Evidence from China Migrants Dynamic Survey"

_ijerph, 2022, doi:10.3390/ijerph19105868_

Round 1
Reviewer 1 Report
The object of study of this research is socially and scientifically relevant, the impact and effects of self-employment on health is a topic that has interested the scientific community and is key to current public policies. The paper analyzes some new and interesting variants in the study of self-employment (inequalities between workers and the most disadvantaged groups of workers). The theoretical framework is adequate and the hypothetical approach is plausible. Introducing gender discrimination and ethnic minorities is an aspect to be highlighted. The sample, which was drawn from the China Dynamic Migrant Survey, is very large and representative. The analyzes carried out are rigorous and correct. The conclusions and implications drawn from the study are relevant and may serve to expand knowledge on the topic and improve the health conditions of the group of workers studied. It is requested that the section on the sampling techniques used in the study and the selection criteria for households and workers be more fully detailed.
Specific Comments:
Please specify in greater detail and depth why this type of sampling technique was used (justification) and how the selection and rejection process of the households and workers who participated in the study was.
Please increase the detail of the technical process when selecting the sampling techniques (procedure, selection and rejection criteria).
Author Response
Dear reviewer,
We sincerely thank the reviewer for thoroughly examining our manuscript and providing very helpful comments to guide our revision. We have uploaded the revised manuscript as an attachment and hope that this revised manuscript has addressed all your comments and suggestions.
Best regards,
Wanting Huang

Reviewer 2 Report
Congratulations to the authors for the effort and approach of an important topic in the economic and labor market reality not only in China, but all over the world.
Remarks:
- the paper requires minor revisions related to writing, missing or extra letters (eg line 32, line 45, line 340, 391, etc.)
- regarding the English language, the authors decide whether to use American English (ex - labor) or British English ( labour), attention to the syntax and grammar.
Suggestions:
- I propose replacing the expression "weaker group" with "vulnerable group" given the nature of the group and its appreciation.
- The theoretical concepts of "psychological identity" and "psyhological identity of urban integration" could be presented in more detail in the paper to emphasize the importance given to it both in Figure 1 and at the end of the paper.
I wish success to the authors in all of their works.
Author Response
Dear reviewer,
We sincerely thank the reviewer for thoroughly examining our manuscript and providing very helpful comments to guide our revision. We have uploaded the revised manuscript as an attachment and sincerely hope that this revised manuscript has addressed all your comments and suggestions.
Best regards,
Wanting Huang

Reviewer 3 Report
This manuscript takes migrant workers in China as an example to explore the impact of self-employment on health, which is an interesting research question. But I think there are still some areas worthy of improvement. I provide the following comments, hoping to be valuable to the author.
Please consider the impact of industry differences. Self-employment does not mean they are in the same industry. People in different industries often face different working environments. For example, workers on construction sites suffer more dust pollution than those in the office. This is an obvious factor affecting health.
In Line 326-327, the the manuscript refers to “six symptoms such as diarrhea, cold and fever are calculated as the morbidity rate”. I'm curious about how this is calculated, because the severity of these diseases is different. Please supplement "morbidity rate" about the sample for each disease.
Author Response

(The authors gave the same response as above.)
